# Development of the Troponin Detection System Based on the Nanostructure

**DOI:** 10.3390/mi10030203

**Published:** 2019-03-22

**Authors:** Taek Lee, Jae-Hyuk Ahn, Jinha Choi, Yeonju Lee, Jin-Myung Kim, Chulhwan Park, Hongje Jang, Tae-Hyung Kim, Min-Ho Lee

**Affiliations:** 1Department of Chemical Engineering, Kwangwoon University, Wolgye-dong, Nowon-gu, Seoul 01899, Korea; mlpad@naver.com (Y.L.); wls629@icloud.com (J.-M.K.); chpark@kw.ac.kr (C.P.); 2Department of Electronic Engineering, Kwangwoon University, Wolgye-dong, Nowon-gu, Seoul 01899, Korea; jaehahn@kw.ac.kr; 3Department of Chemical and Biomolecular Engineering, Sogang University, 35 Baekbeom-ro (Sinsu-dong), Mapo-gu, Seoul 04107, Korea; jinhachoi2@gmail.com; 4Department of Chemistry, Kwangwoon University, Wolgye-dong, Nowon-gu, Seoul 01899, Korea; hjang@kw.ac.kr; 5School of Integrative Engineering, Chung-Ang University, Heukseok-dong, Dongjak-gu, Seoul 06974, Korea; thkim@cau.ac.kr

**Keywords:** cardiac troponin detection system, acute myocardial infarction, electrochemical, field effect transistor, surface plasmon resonance, surface enhanced Raman spectroscopy, nanostructure

## Abstract

During the last 30 years, the World Health Organization (WHO) reported a gradual increase in the number of patients with cardiovascular disease (CVD), not only in developed but also in developing countries. In particular, acute myocardial infarction (AMI) is one of the severe CVDs because of the high death rate, damage to the body, and various complications. During these harmful effects, rapid diagnosis of AMI is key for saving patients with CVD in an emergency. The prompt diagnosis and proper treatment of patients with AMI are important to increase the survival rate of these patients. To treat patients with AMI quickly, detection of a CVD biomarker at an ultra-low concentration is essential. Cardiac troponins (cTNs), cardiac myoglobin (cMB), and creatine kinase MB are typical biomarkers for AMI detection. An increase in the levels of those biomarkers in blood implies damage to cardiomyocytes and thus is related to AMI progression. In particular, cTNs are regarded as a gold standard biomarker for AMI diagnosis. The conventional TN detection system for detection of AMI requires long measurement time and is labor-intensive and tedious. Therefore, the demand for sensitive and selective TN detection techniques is increasing at present. To meet this demand, several approaches and methods have been applied to develop a TN detection system based on a nanostructure. In the present review, the authors reviewed recent advances in TN biosensors with a focus on four detection systems: (1) An electrochemical (EC) TN nanobiosensor, (2) field effect transistor (FET)-based TN nanobiosensor, (3) surface plasmon resonance (SPR)-based TN nanobiosensor and (4) surface enhanced Raman spectroscopy (SERS)-based TN nanobiosensor.

## 1. Introduction

As technologies have been developed, human health and welfare improve through adequate medical care and services that extend the lifespan and health-span as compared to those of past generations. Nevertheless, humans are growing old and this process leads to several diseases [1,2]. Those diseases and illnesses prevent healthy life in old age. Especially cardiovascular diseases (CVDs), which are a serious threat to human life because of the high death rate, difficult recovery, and its influence on other organs [3,4]. In 2012, the World Health Organization (WHO) reported that 31% (17.5 million) of deaths all over the world were attributable to patients with CVDs [5]. When people age without balanced nutrition and sufficient physical exercise, obviously, the incidence of CVDs increases in developed countries. In addition, in the case of developing countries, unhealthy conditions such as poor air quality, emotional stress, unbalanced nutrition, and insufficient medical services drastically increase the incidence of CVDs [6].

Among CVDs, myocardial infraction (MI) is one of the dangerous diseases because of the high death rate and its impact on other organs. MI can be subdivided into chronic myocardial infraction and acute myocardial infraction (AMI) [7]. Both diseases originate from the blockage of a coronary artery by a blood clot or formation of an atherosclerotic plaque in the coronary artery, or by other problems. These phenomena interrupt the blood flow, thus inhibiting the oxygen and nutrient transfer between the heart and blood vessels; this situation damages myocardial cells on the heart surface. These phenomena increase in incidence with atherosclerosis incidence [8]. In particular, AMI is strongly related to heart attacks and cardiac arrest, whose death rate increases without rapid and adequate treatment in an emergency. Accordingly, AMI requires rapid diagnosis and treatment to save a patient’s life efficiently. Most emergency departments say that the turnaround time for detecting AMI is within 60 min, but 30 min is ideal for treatment of AMI patients to minimize organ damage [9,10,11]. To achieve this goal, the early and precise detection of AMI is key. At present, several biomarkers are used to detect AMI, e.g., cardiac troponins (cTNs), cardiac myoglobin (cMB), and creatine kinase MB (CK-MB) in blood tests [8,12]. Only, a simple blood-based AMI test is available when AMI occurs. In particular, troponins (TNs: TNI and TNT) are regarded as the gold standard for AMI detection because of a correlation between AMI and TN levels in blood [13]. Various diagnostic methods have been proposed to detect AMI. For example, electrocardiography [14] and radiology-based techniques [15,16] provide enough information for AMI detection. Nonetheless, these assays are hard to perform in an emergency. For emergencies, rapid detection of a TN is required. In particular, developers of TN detection systems have recently focused on portability, small sample consumption, short detection time, and high accuracy for construction of a point-of-care test (POCT) system.

To date, various types of TN detection platforms have been developed, for example, an enzyme-linked immunosorbent assay (ELISA) [17], surface plasmon resonance (SPR) [18], radioimmunoassay (RIA) [19], surface-enhanced Raman spectroscopy (SERS) [20], fluorescence [21], electrochemical (EC) method [22], and the field effect transistor (FET)-based detection method [23]. Among them, EC, FET, SPR, and SERS-based TN detection methods have received attention for POCT development. Especially as these techniques can be easily tuned with their functionality when the nanostructure or nanoparticle is introduced such as sensitivity, selectivity, and enabling the device for operation [18,20,22,23]. For constructing a POCT system for AMI diagnosis, a novel diagnostic tool with a nanostructure is urgently needed [24,25,26]. To shed light on the current progress of nanostructure-based TN detection systems, in this paper, we reviewed the recent advances in four types of TN detection systems based on a nanostructure, including EC, FET, SPR, and SERS-based methods.

## 2. EC-Based TN Detection System with Nanostructure

For detecting a TN in an emergency, the most important factors are detection time and concentration of the TN (~1 pM) within the clinically important period (~1 h). The EC TN detection method is suitable for those requirements [13,22,27]. The EC TN detection method offers portability (electrode size miniaturized through nano/microtechnology), modularity (can be integrated with a smartphone or arduino-based device), minimal requirements for sample volume, and short measurement time [28,29,30]. Usually, EC TN detection can be achieved by such methods as voltammetric, potentiometric, conductometric, amperometric, and impedimetric techniques [31,32,33]. The potential and current change by a redox material–mediated reaction between a target TN and a bioprobe can be used to determine the TN concentration accurately [34]. In addition, the resistance change between the target (analyte) and bioprobe gives clues to a binding event in the fabricated biosensor substrate [35]. For constructing a TN biosensor, EC biosensors can be classified into two types of bioprobe: An immunosensor (antibody-mediated bioprobe) [35,36] and aptasensor (aptamer-mediated bioprobe) [29,34]. In particular, when a nanostructure is combined with an EC TN detection method, several advantages such as a better limit of detection (LOD), small loaded sample, and assay specificity can be gained. Because the nanostructure provides a greater binding chance between the bioprobe and electrode compared to the normal substrate. The use of a nanostructure (e.g., nanoparticle, nanocomposite, or nanotube), including on the electrode, improves the surface roughness, reactivity, bioprobe immobilization efficiency, and electron transfer between a target and the bioprobe-modified electrode [13,36,37].

Recently, Lee’s group fabricated an EC-based TNI biosensor composed of multifunctional DNA and Au nanospikes on an Au microgap with a printed circuit board chip [38] (Figure 1A). As a bioprobe, a DNA three-way-junction (3WJ) was introduced because the DNA 3WJ has three arms for multi-functionality. Each fragment of DNA 3WJ was rolled into a recognition part (cTNI detection aptamer), an EC signal transduction part (methylene blue), and an immobilization part (thiol group). Each piece of DNA was assembled to form the DNA 3WJ for TNI detection, signal transduction, and immobilization simultaneously. Moreover, to increase the EC signal sensitivity, an Au nanospike (AuNS) was prepared. The Au microgap array was integrated with a printed circuit board chip to control each microgap electrode panel selectively to detect a small amount of a TN. A cyclic voltammetry (CV) experiment was conducted to confirm the binding of the TN to the DNA 3WJ-modified electrode. This TNI biosensor had a LOD of 1 pM in a HEPES (2-[4-(2-hydroxyethyl)piperazin-1-yl]ethanesulfonic acid) buffer and 1 pM in human serum, respectively (Figure 1B). In addition, a specificity test was carried out with other proteins including hemocyanin, myoglobin, hemoglobin, and albumin (Figure 1C). They tried to apply the multifunctional DNA structure and AuNS to a portable label-free EC biosensor.

Xiong et al. recently fabricated an EC impedance spectroscopy (EIS)-based immunosensor composed of an anti-cTNI antibody and Ag nanoparticles (AgNPs) [35]. AgNPs were synthesized from green algae Stoechospermum marginatum through biosynthesis. The biosynthesis of AgNPs was carried out from an Ag salt and algal extract. For the biosensor fabrication, a self-assembly method was employed. 3-Aminopropyl triethoxysilane was attached to an indium tin oxide (ITO) substrate. Then, the AgNPs (with mercaptopropionic acid (MPA)-N-hydroxysuccinimide (NHS)) were 3-aminopropyl triethoxysilane–conjugated to ITO through a coupling reaction. Next, an anti-cTNI antibody was immobilized on the modified substrate. For preventing physical-adsorption immobilization, bovine serum albumin (BSA) was applied to the antibody/AgNP/ITO electrode. Ferricyanide was introduced to obtain a change in the redox property between the target and bioprobe during the reaction. The EIS experiment was conducted to confirm the electrode fabrication. Furthermore, when the target Ag-cTNI was added at 0.02 to 1.00 μg/mL, faradaic impedance spectra showed a high correlation between the analyte concentration and electron transfer resistance. They reported that the LOD was 0.001 μg/mL, which meets the value for AMI determination.

Furthermore, the Dutra group has proposed an EC cTNT immunosensor composed of an antibody and a carbon nanotube (CNT) using amperometry [39]. For constructing the biosensor electrode, polyethyleneimine (PEI) was applied to a bare Au substrate. Then, carboxylated multi-walled carbon nanotubes (COOH-MWCNTs) were immobilized with 1-Ethyl-3-(3-dimethylaminopropyl)-carbodiimide (EDC)/NHS treatment. After that, an anti-cTNT antibody was immobilized onto the CNT-modified substrate with glycine treatment for blocking. The CV was performed during the nanobiofilm fabrication. To identify optimal biosensor fabrication conditions, they investigated the effects of various PEI concentrations and various CNT concentrations via CV with hydrogen peroxide and potassium ferricyanide, respectively. Additionally, pH and PBS concentration effects were determined to find the optimal conditions. They stated that the optimal conditions in terms of PEI, CNT, pH, and PBS are 5%, 2 mg/mL, and 7 and 10 mmol, respectively, for the creation of a cTNT immunosensor. The fabricated biosensor can detect the cTNT concentration from 0.1 to 1.0 ng/mL with a linear relation. The LOD was found to be 0.033 ng/mL. Moreover, several EC biosensors were devised to detect a TN with nanostructures including gold nanoparticles, gold nanodumbbells, ZnO nanostructures, and a CNT-based screen-printed electrode [40,41,42,43]. A clinical test was performed on human serum samples, and the results were compared with those of an electrochemiluminescence method [44]. Thus, a nanostructure can provide various platforms for fabricating an EC TN biosensor effectively.

## 3. FET-Based TN Detection System with Nanostructure

In a typical FET-based biosensor, a semiconductor channel material is connected between two closely spaced metal electrodes, and the channel material is functionalized with receptors such as an antibody or aptamer to capture the target analyte. High sensitivity can be achieved by means of nanomaterials including silicon nanowires (SiNWs) [45,46,47,48], CNTs [49,50], graphene [51,52], and MoS_2_ [53], which have a high surface-to-volume ratio compared to a micro-structured electrode. Those nanostructures provided the platform to fabricate the FET-devices. Analysis time for FET-based biosensors is tremendously decreased due to direct electrical detection without an additional labeling process. This simple and rapid diagnostic method is well suitable for detection of a cardiac marker (i.e., a TN) of AMI where early diagnosis and fast treatment are required.

Kim et al. have demonstrated a SiNW FET with a honeycomb-like structure for detection of a TN [54]. Compared to conventional straight-line nanowires, the honeycomb-like structure has a larger effective channel width, and thus the output signal change is improved. The honeycomb-like structure was fabricated by electron beam lithography and reactive ion etching. An Ag/AgCl pseudo-reference electrode was embedded in the chip to serve as the liquid-gate electrode. Monoclonal antibodies against cTNI were immobilized on the surface of the nanowires. The LOD was determined to ~5 pg/mL in PBS with the specificity unaffected by C-reactive protein (CRP) and α-fetoprotein (AFP).

Liu et al. have presented a low-cost, time-efficient, and scalable lithography-free method for fabrication of In_2_O_3_ nanoribbon biosensor arrays. Two simple shadow masks combined with a lift-off process were employed to define and pattern the nanoribbons and metal electrodes [55]. A sandwiched structure with a biotinylated secondary antibody was utilized to bind a biotinylated urease through streptavidin. The sandwiched structure can be considered an electronic enzyme-linked immunosorbent assay (ELISA) technique where the change in pH caused by the urease enzymatic activity is the amplification signal instead of direct electrical detection of biomolecules. With the LOD of 1 pg/mL in PBS, the In_2_O_3_ nanoribbon biosensor could detect added TN (down to 10 pg/mL) in diluted human whole blood.

The advantage of a FET-based biosensor is the easy integration with electronic circuits suitable for portable application. Livi et al. have demonstrated monolithic integration of an array of SiNW FETs with signal-conditioning circuits on the same chip [56]. After the synthesis by chemical vapor deposition, SiNWs were transferred to a complementary metal–oxide–semiconductor (CMOS) chip and then electrically connected to metal contacts by e-beam lithography and a lift-off process. Signal outputs generated from the multiple SiNWs were processed on a chip and then transmitted to a receiving unit (i.e., a host PC). The differences in nanowire responses caused by the variation of the intrinsic electrical characteristics were calibrated by means of the initial drain currents to provide uniform electrical outputs. The LOD was 1 nM cTNT in phosphate buffer achieved from an averaged calibrated response measured for 10 individual SiNWs.

One of the critical challenges for practical application of FET-based biosensors to physiological conditions is the charge-screening effect at high ionic strength in solution [57]. Although a diluted buffer is used to increase the Debye screening length, the diluted solution may degrade the protein activity and binding affinity. Agarwal et al. have demonstrated a ZnO thin film transistor functionalized with DNA aptamers for detection of cTNT (10 μg/mL) [58]. Target-induced conformational changes of aptamers close to semiconductor channels can modulate channel conductance in physiological buffers [59]. The functionalization was confirmed by analyzing the surface with atomic force microscopy. The Kelvin probe force microscopy technique revealed that the surface potential difference originated from a specific interaction between the aptamer and cTNT. Recently, a novel FET-based biosensor was demonstrated to overcome the Debye screening effect by means of an electric-double layer (EDL) for the detection of biomolecules in a high-ionic-strength solution [60]. Sarangadharan et al. have presented an EDL-gated AlGaN/GaN high–electron mobility transistor for the detection of a TN in clinical serum samples [61]. The sensing region was composed of an open gate electrode functionalized with TN receptors (antibody or aptamer) and the open transistor channel (Figure 2A). Figure 2B shows that the net potential drop (*V_g_*) across the solution is the sum of potential drops in the EDL formed at the gate electrode (∆*V_s_*) and the transistor channel (∆*V_ox_*), expressed as $\Delta V_{ox}=\frac{C_{s}}{C_{ox}+C_{s}}V_{g}$. The gate voltage drop in a dielectric (∆*V_ox_*) is affected by the solution capacitance (*C_s_*) formed by EDL, expressed as where Cox is dielectric capacitance. Receptor–ligand interaction on the gate electrode can result in a change in *C_s_* and ∆*V_ox_* and, thus, in the channel current (Figure 2C). Based on this working mechanism, the EDL biosensor could detect the TN in a wide dynamic range (0.006–148 ng/mL) in clinical human serum samples without an additional wash or sample pretreatment step (Figure 2D).

As discussed above, nanoscale FET-based electrical detection of cTNs has a low LOD with fast response time even in clinical samples; this configuration is suitable for early diagnosis of CVDs. Wearable-type FET-based biosensors can be implemented using flexible nanomaterials [62]. The continuous monitoring of a CVD is expected to be achieved with the wearable biosensors in a minimally invasive form in addition to the development of a new type of cardiovascular biomarker for nonblood body fluids such as sweat, tear fluid, and urine. Table 1 shows the comparison of EC-based TN biosensor and FET-based TN biosensor in terms of several factors.

## 4. SPR-Based TN Detection System with Nanostructure

Compared to other optical biosensors, the SPR-based detection method can provide straightforward information about the parameters between a target and a bioprobe at nanoscale biointerfaces without a labeling or amplification process [63,64,65]. An SPR-based biosensor can detect a binding event between a target such as a protein, nucleic acid, or a small molecule and a bioprobe including an antibody, aptamer, or nanoparticles by monitoring the refractive changes on the metallic surface [66,67]. The refractive index change originates from the application of light to the metallic surface; this process causes electron oscillation between a free electron and the charged metal surface with the surface plasmon effect. Because of simplicity, short detection time, portability, and cheapness of the equipment, an SPR-based biosensor can be easily implemented in various disease detection systems [68,69,70]. In particular, the SPR technique has emerged in nanobiotechnology and extends the applications of SPR-based biosensors to field-ready POCT devices [71,72,73]. TN detection by SPR is one of the interesting applications for AMI diagnosis. Normally, the SPR-based TN detection can be classified into two types: An immunosensor and aptasensor [13,37,74]. Usually, the nanostructure has improved LODs and nanostructure orientation can be altered for an SPR effect enhancement, thus shortening the fabrication process and bioprobe development for multi-functionality. Moreover, the LSPR-based biosensor can be fabricated with the nanostructure. Thus far, SPR-based TN biosensors have been developed into an immunosensor.

Recently, Wu et al. created a magnetic-field–assisted SPR biosensor composed of an antibody, hollow gold nanoparticles, and polydopamine (PDA) [75]. For developing a TN biosensor, the capture antibody was immobilized onto a hollow gold nanoparticle+PDA–modified Au substrate. Next, MWCNTs were coated with amine-functionalized Fe3O4 nanoparticles, PDA, and a detection antibody for magnetic SPR application. The newly prepared immunosensor was reacted with TN standard solutions in a magnetic field. The fabricated SPR-based biosensor yielded results within 40 min. By this modified SPR technique, they could detect a TN at ~1.25 ng/mL. Ro group has reported an SPR-chip TN immunosensor using a TN peptide antigen epitope and an anti-TN monoclonal antibody on an Au thin film-modified slide glass [76]. To fabricate the SPR chip, a TN peptide antigen and an anti-TN monoclonal antibody were prepared by recombinant-DNA techniques and a hybridoma cell-based antibody production method, respectively. After that, the fabricated SPR chip was tested by means of the TN epitope. When the SPR experiment was conducted, the intensity of the SPR and concentration of the TN correlated in a range from 0 to 160 ng/mL. The LOD of the proposed TN sensor was 0.068 ng/mL. For a clinical trial, human serum was tested with the fabricated SPR chip, and the method produced an SPR angle shift well. Some groups have focused on the regeneration of TN immunosensors for repeated reuse. A reusable cTNT immunosensor has been proposed that involves gold nanorods (GNRs), an anti-cTNT antibody, and thermosensitive amine-modified poly(N-isopropylacrylamide) (PNIPAAM) [77]. For constructing a reusable cTNT detection system, GNRs were conjugated with the anti-TNT antibody. Then, the amine-modified PNIPAAM was coated with a cTNT/GNR system through an EDC/NHS coupling reaction. UV-near-infrared, Fourier transform infrared, and transmission electron microscopy experiments revealed the formation of PNIPAAM-anti-TNT-GNR (Figure 3A). The regeneration principle of the fabricated localized-SPR (LSPR) cTNT biosensor is thermal treatment after binding of target cTNT. Normally, cTNT was reacted with the PNIPAAM/TNT/GNR structure at 25 °C (λ: 844 nm). When the temperature was changed to 37 °C for 20 min, the dielectric constant of the medium near the TNT-PNIPAAM/TNT/GNR structure changed, thus triggering dissociation of the TNT-PNIPAAM/TNT/GNR complex for regeneration of the LSPR immunosensor (λ: 835 nm; Figure 3B,C). Computational modeling with molecular docking, dynamics analysis, and free-energy calculation revealed an expected target–bioprobe association/dissociation mechanism. Besides, the cTNT detection performance of the fabricated immunosensor showed a wide linear range between 7.6 fg/mL and 910 μg/mL as compared to a general cTNT biosensor. The LOD was 8.4 fg/mL, and the response time was 10 s. That study offers regeneration of an LSPR cTNT immunosensor by a simple temperature change method. Furthermore, Pawula et al. have constructed two reusable types of cTNT immunosensors (direct type and sandwich type) composed of an anti-cTNT antibody and gold nanoparticles with SPR [78]. In addition, Dutra et al. have developed a reusable cTNT biosensor with commercially available SPR AUTOLAB SPIRIT [79]. The antibody-and-GNR hybrid nanostructure has been implemented [80], and an anti-fouling antibody has been employed to detect cTNI by an SPR technique [81]. Furthermore, a fiber optics-based SPR biosensor has been developed to detect cTNI and MB [82]. In addition, the peptide-functionalized gold nanoparticles were applied to detect troponin [83] and the exploiting surface-plasmon-enhanced light scattering method was introduced [84].

## 5. SERS-Based TN Detection System with Nanostructure

Raman spectroscopy is a noninvasive analytical method for determining the chemical identity and state of biomolecules via their distinctive scattering properties [85,86,87]. A Raman shift means that the wavelength of inelastic scattering of biomolecules can be shifted because the energy of the scattered photons depends on the chemical properties of each molecule. Specific Raman shift peaks from analytes can be characterized as a spectrum or images for identification of an unknown sample. By this analytical method, biomarkers can be quantified in a precise manner. Nevertheless, Raman intensity is too weak to determine the low concentration of various biomarkers, which require sensitive detection for the exact diagnosis of diseases. On the other hand, SERS is a signal-enhancing technique exploiting plasmon effects on a metal surface for quantitating biomolecules sensitively, as compared to conventional Raman spectroscopy [88,89,90,91,92]. When the well-ordered nanostructure was introduced to the substrate, then, enhanced Raman peaks showed the fingerprint of target molecular and bioprobe. For detection of some biomarkers, metallic nanostructures are usually used to induce SERS effects where the electromagnetic fields related to plasmon modes are employed to increase Raman vibrational signals from the biomolecules [91]. With possible enhancement factors in the order of 1014, SERS is an exceptionally sensitive detection method. Recent developments in nanotechnology enabling sophisticated nanostructures, have contributed to the popularity of SERS for the early diagnosis and management of diseases where biomarker detection is prevalent [93,94,95]. For example, special attention has been given to SERS-based immunoassays mediated by antigen–antibody binding. Although SERS may provide high sensitivity for target molecules, the specific interaction within an antibody–antigen pair offers highly selective detection. Consequently, a combination of the two ensures that SERS-based biosensor platforms are suitable for detection of low-concentration biomarkers for early disease diagnosis. In this section, recent studies related to SERS-based TN biosensors are briefly introduced.

Au nanoparticles have been frequently utilized to induce the SERS effect in a TN immunoassay system. As mentioned above, the surface of a noble metal offers surface plasmons and could enhance Raman intensity, especially on a Raman-active probe such as malachite green isothiocyanate or X-rhodamine-5-(and-6)-isothiocyanate. One research group has successfully developed a SERS-based immunoassay using malachite green isothiocyanate- or X-rhodamine-5-(and-6)-isothiocyanate–coated hollow gold nanospheres [96]. Figure 4A presents the immunoassay process for simultaneous detection of a TN and CK-MB, which are some of the most representative cardiac markers. Target antibodies for the capture of TN and CK-MB simultaneously were immobilized on magnetic microbeads, and target molecules (TN and CK-MB) were conjugated to the surface of the hollow gold nanospheres with a distinct Raman-active dye. Via this strategy, different analytes could be selectively quantified in a one-pot reaction, although those authors implemented the method for blood serum at a single excitation wavelength. The authors claimed that this approach does not require sample preparation, such as centrifugation or filtration, and is significantly less influenced by sample dilution and matrix effects than the sandwich immunoassay format is. Another group has employed Ag−Au nanostars and a three-dimensional ordered macroporous (3DOM) Au−Ag−Au plasmonic array for the sensitive and selective detection of TN, N-terminal prohormone of brain natriuretic peptide (NTProBNP), and neutrophil gelatinase-associated lipocalin (NGAL) for the early diagnosis of cardiorenal syndrome through induction of the SERS effect, serving as a “hot field” [97]. In Figure 4B, Raman dye-decorated Ag–Au nanostars and a 3DOM plasmonic array are characterized by an analytical method involving scanning electron microscopy. The shape of the nanostar and porous structure helped to enhance Raman intensity and improved sensitivity for the analyte owing to the formation of a hot field between the nanostar and 3DOM array if the target bound to them. The authors reported that the LODs are 0.76, 0.53, and 0.41 fg/mL for TN, NT-ProBNP, and NGAL, respectively. Fu et al. have also demonstrated a signal amplification SERS platform for the detection of a TN by means of gold nanoparticles, graphene oxide (GO), and magnetic beads [98]. The GO–gold nanoparticle complex played a dual role: To amplify the SERS signal and target the TN, which was captured by the antibody-functionalized magnetic beads. TN concentration was successfully measured in a highly sensitive manner, in a range from 5 pg/mL to 1000 ng/mL. On the other hand, it was difficult to quantify TN at concentrations lower than 1 ng/mL without GO. That study suggests that the gold nanoparticle-GO complex has a synergistic positive effect on SERS intensity for sensitive detection. Garza et al. have created a unique collection device and method for a consistent and precise signal corresponding to an analyte concentration by collecting Raman dye–decorated nanoprobes via vacuum [99]. When this collecting and measuring system is employed, several disadvantages from the nanoprobes in solution may manifest themselves such as a reduction in sensitivity and heterogeneity of the signals. Different concentrations of the nanoprobe were measured on the device, and its LOD was 12.9 fM. Moreover, a TN was detected at a concentration as low as 1 ng/mL, whereas a noncollecting device could not detect 100 ng/mL. These data suggest that the nanoprobe-collecting strategy improves the sensitivity of target detection. In addition, the coefficient of variation was found to be less than 10% for the medium of the nanoprobe concentrations tested. This means that the nanoprobe-collecting strategy may also be exploited to improve reproducibility. Table 2 shows the comparison of SPR-based TN biosensor and SERS-based TN biosensor in terms of several factors.

Recently, a lateral flow immunoassay was applied to detect TNs for rapid, simple detection and POCT. A SERS-based TN immunoassay has also been integrated into the lateral flow immunoassay. For example, one research group has created a single-line multiplexed assay for a TN, MB, and C-reactive protein (CRP), with a range from pg/mL to µg/mL, by functionalizing distinguishable Raman nanoprobes (highlighted at 923, 1160, and 1335 cm^-1^) with appropriate antibodies for each analyte [100]. On this platform, the TN was successfully measured in a 15 μL sample in the presence or absence of CRP. Zhang et al. have devised a core–shell SERS nanotag-based multiplex lateral flow immunoassay for rapid and quantitative detection of three cardiac biomarkers, MB, TN, and CK-MB [101]. A silver-gold core–shell nanoparticle with Nile blue A encapsulated in the interior gap between the two metals was used in the conjugate pad. Detection antibodies for the three biomarkers were conjugated with respective SERS nanotags, and three test lines were fabricated on a nitrocellulose membrane for multiplex detection. After the flow of a sample from the sample pad to absorption pad, Raman signals of the three test lines were measured for the quantification of cardiac biomarkers. The LODs for MB, TN, and CK-MB were found to be below the clinical cutoff values, which were 3.2, 0.44, and 0.55 pg/mL, respectively. The dynamic ranges for MB, TN, and CK-MB were 0.01−500.00, 0.01−50.00, and 0.02−90.00 ng/mL, respectively, which covers the clinical range. In the same research group, a single–test line multiplex lateral flow immunoassay for quick diagnosis of AMI has been developed [102]. In that study, the authors utilized a single test line for the measurement of three analytes related to AMI and successfully detected them simultaneously (CK-MB, TN, and MB quantitation ranges were 0.02−90.00, 0.01−50.00, and 0.01−500.00 ng/mL, respectively). The advantages compared to the use of three test lines are a reduction in preparation procedures, lower reagent consumption, and cost, decreased preparation time, and easy experiment operation.

## 6. Outlook

As the income level of humans gradually improves in developing and developed countries, the quality of healthcare services should improve to prevent, diagnose, and treat injuries, disorders, and diseases. Especially, the diagnosis of AMI during the early stages is essential for saving a patient’s life and for their adequate treatment and recovery. TNs are regarded as some of the reliable biomarkers for AMI diagnosis. Rapid detection of a TN with high accuracy, small volume, and at an ultra-low concentration (lower than 24 pg/mL in a blood sample) is essential for fabricating a POCT TN detection system. Thus far, various biosensors and equipment have been proposed to detect TNs. Nevertheless, those platforms hardly meet the requirements for constructing a handheld TN detection kit, due to the involvement of multiple detection methods, complicated equipment, and long detection time. As we discussed in this review, introducing a nanostructure may improve biosensor performance in terms of sensitivity, specificity, fidelity, required volume, and detection time. These nanostructures can be easily tailored to integrate the functionality and for modularity with miniaturized devices. Up until now, integrating a nanostructure into a biosensor has been somewhat successful as various types of biosensors based on a nanostructure have been developed to overcome the current limitations of biosensors. It is obvious that the advantages of nanostructures for TN biosensor construction include (1) the nanostructure-based electrode substrate can enhance the signal sensitivity and specificity for EC, FET, SPR, and SERS-based biosensors; (2) because of the sensitivity increase, a lesser volume of sample is required as compared to a traditional biosensor; (3) introducing the nanomaterial might enable new detection techniques or reduce the detection time for optical biosensors. Furthermore, it is hard to operate the LSPR and SERS measurements without nanostructure. Like this, introducing the nanostructure has extended ways for troponin detection methods. The present review addressed four types of biosensor (EC, FET, SPR, and SERS) involving a nanostructure. Nevertheless, there is no commercial product for the detection of a cTN using a nanobiosensor. Our next challenge in the field of nanostructure-based biosensors for AMI diagnosis will be a POCT measurement, compatibility with a smartphone module, quality control of bio-probe mass production, and uniform fabrication of nanostructures for manufacturing a TN detection kit.

## Figures and Tables

**Figure 1 micromachines-10-00203-f001:**
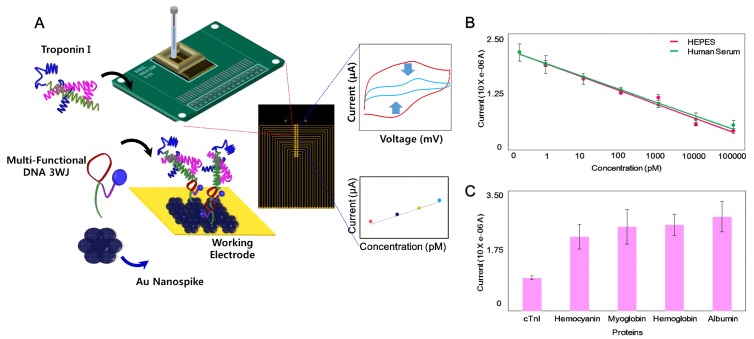
(**A**) Schematic diagram of the electrochemical (EC)-based troponin (TNI) biosensor consisted of DNA 3WJ/Au nanospike on Au micro-gap/PCB system, (**B**) reduction potential of the fabricated biosensor to various concentrations of TNI in HEPES (2-[4-(2-hydroxyethyl)piperazin-1-yl]ethanesulfonic acid) buffer and human serum (0 pM to 100 nM), (**C**) reduction potential of the fabricated biosensor to various target proteins in the HEPES buffer (TNI, Hemocyanin, Myoglobin, Hemoglobin, Albumin). (Reproduced with permission from the authors of reference [38], published by Elsevier, 2019).

**Figure 2 micromachines-10-00203-f002:**
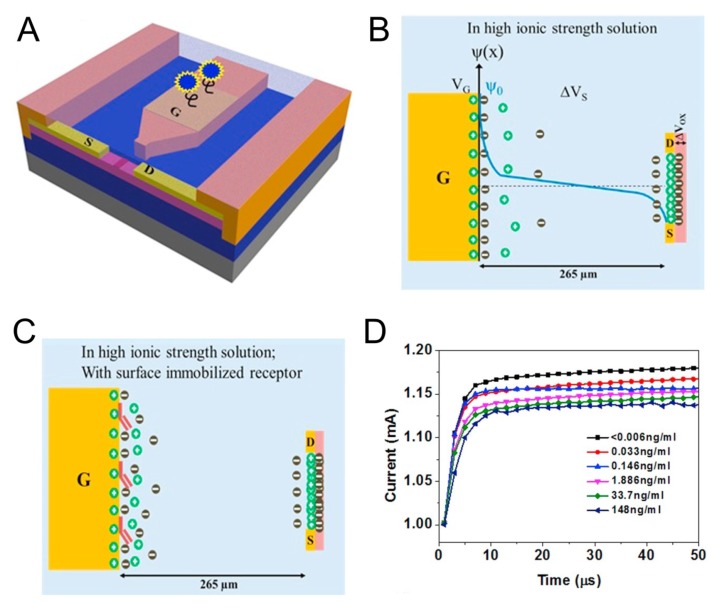
(**A**) Schematic illustration of the AlGaN/GaN High Electron Mobility Transistor (HEMT) sensor consisting of a gate electrode opening and channel opening separated by a fixed distance. (**B**) Potential distribution across the solution when gate voltage and transistor bias are applied. (**C**) Illustration of the charge distribution in the electric-double layer (EDL) gated HEMT structure when the receptor is immobilized on the gate electrode area. (**D**) Current versus time graph for the aptamer based detection of TNI in clinical human serum samples [61].

**Figure 3 micromachines-10-00203-f003:**
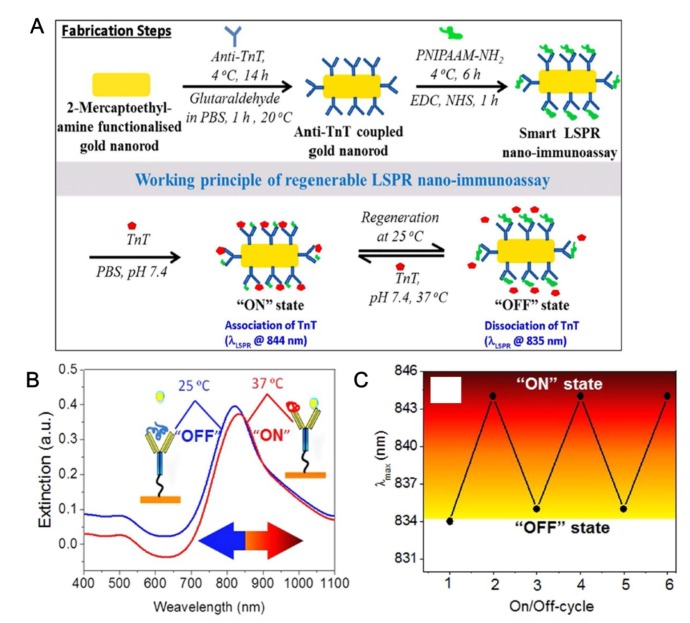
(**A**) Schematic presentation of fabrication steps involved with regenerable LSPR nano-immunoassay and its reversible working mode of action at 25 and 37 °C. (**B**) UV-NIR spectra of GNR-anti-TnT-PNIPAAM LSPR nano-immunoassay when 5 ng/mL TnT solution was treated at 37 °C (associated) and 25 °C (dissociated). (**C**) Regeneration of the GNR-anti-TnTPNIPAAM LSPR nano-immunoassay at 37 °C and 25 °C when a 5 ng/mL solution was used. (Reproduced with permission from the authors of reference [77], the figure has followed the terms of use under a Creative Commons Attribution 4.0 International License.).

**Figure 4 micromachines-10-00203-f004:**
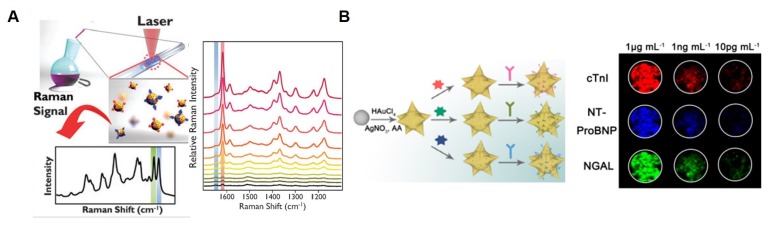
SERS-based immunoassay biosensors using nanoparticles for sensitive detection. (**A**) Competitive immunoassay of TN and CK-MB markers using magnetic microparticles and gold nanoparticles. Reproduced with permission from [96]. Copyright 2014, Royal Society of Chemistry. (**B**) Bimetallic (gold and silver) nanostars and gold-silver nanoarray for inducing of the hot field related to the SERS effect and highly sensitive detection of the TN. Reproduced with permission from [97]. Copyright 2018, ACS Publications.

**Table 1 micromachines-10-00203-t001:** Comparison of the electrochemical (EC)-based biosensor and field-effect transistor (FET)-based for TN detection in terms of the bioprobe, detection method, detection limit, and nanostructure.

Bioprobe	Detection Method	Detection Limit	Nanostructure	Ref
Antibody	CV/EIS	0.2 ng/mL	Carbon nanofiber	[27]
Antibody	CV/EIS	24 pg/mL	Gold nanoparticle	[34]
Aptamer	CA	24 pg/mL (1 pM)	Fc-modified silica nanoparticle	[29]
Aptamer	DPV	8 pg/mL	Au nanodumbbells	[41]
Aptamer	CV	24 pg/mL (1 pM) (in a dilluted serum)	Au nanospike	[38]
Antibody	Direct electrical detection	5 pg/mL (cTnI)	Silicon nanowires	[54]
Antibody	Sandwich immunoassay, Electrical detection	1 pg/mL (cTnI)	Indium oxide (In_2_O_3_) Nanoribbons	[55]
Antibody	Direct electrical detection	1 nM (cTnT)	Silicon nanowires	[56]
Aptamer	Direct electrical detection	10 μg/mL (cTnT)	Zinc oxide (ZnO) thin film	[58]
Antibody, Aptamer	Electric-double layer, Direct electrical detection	6 pg/mL (cTnI)	AlGaN/GaN nanoribbons	[61]

**Table 2 micromachines-10-00203-t002:** Comparison of the SPR-based TN biosensor and SERS-based TN biosensor in terms of the bioprobe, detection method, detection limit, and nanostructure.

Bioprobe	Detection Method	Detection Limit	Nanostructure	Ref
Antibody	SPR	1.25 ng/mL	Magnetic multi-walled carbon nano-tubes(MMWCNTs)/Hollow gold nanoparticles(HGNPs)	[75]
Antibody	SPR	68 ng/L	-	[76]
Antibody	LSPR	7.6 fg/mL	Gold nanorod	[77]
Antibody	SPR	0.5 ng/mL	Gold nanoparticle	[78]
Antibody	SPR	0.05 ng/mL	-	[79]
Antibody	SPR	100 ng/mL	Gold nanorod	[80]
Antibody	SERS	33.7 pg/mL	Magnetic microparticle/Gold nanoparticle	[94]
Antibody	SERS	0.76 pg/Ml	Bimetallic nanostar (gold-silver)/Gold-silver nanoarray	[95]
Antibody	SERS	5 pg/mL	Graphene oxide/Gold nanoparticle/Magnetic microparticle	[96]
Antibody	SERS	12.9 fM	Magnetic microparticle/Silver nanoparticle	[97]
Antibody	Lateral immunoassay, SERS	1 ng/mL	Gold nanoparticle	[98]
Antibody	Lateral immunoassay, SERS	0.44 pg/mL	Silver-gold core-shell nanoparticle	[99]
Antibody	Lateral immunoassay, SERS	0.89 pg/mL	Silver-gold core-shell nanoparticle	[100]

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
