# Peer review of "Development of the Troponin Detection System Based on the Nanostructure"

_micromachines, 2019, doi:10.3390/mi10030203_

Round 1
Reviewer 1 Report
The manuscript submitted by Lee et. al. summarizes the development of the troponin detection system using nanostructures. Overall, the manuscript is well written and includes the four major detection system of cardiac troponin. The reviewer suggests minor revision of this manuscript before its publication.
Minor revisions:
1. The reviewer suggests to remove Table 1 to p7, line 256 as the authors compare the EC-based and FET-based techniques. So, it would be more convenient for the readers to be better understanding the differences of these techniques by putting the comparison at the end of FET section.
2. The concentration description needs to be consistent throughout the manuscript. Either with ng/mL or ngmL-1.
3. A white square in figure-3. Please modify it.
4. Regarding the surface plamon resonance based technique for the detection of cardiac troponin I, there are two recent published nice papers to be included in this review article for comparison. (ACS Sens. 2016, 1, 12, 1416-1422; Anal. Chem. 2016, 88, 23, 11924-11930).
5. The reference section is messy. Missing pages in references: 2, 7, 12, 21, 45, 46, 60, 63, 69, 71, 77, and 88. Some references use journal full name but some are with abbreviation. Please refer to the journal guidelines.
6. Typos: p3, line 129 (TNi); p5, 198 ([55Liu]); p6, line 246 (TN I); p9, line 379-380 (concentration units); p10, Table 2, ref 95 (0.76 pg/Ml); p10, 402-403, (concentration and wavelength units).
Author Response
The manuscript submitted by Lee et. al. summarizes the development of the troponin detection system using nanostructures. Overall, the manuscript is well written and includes the four major detection system of cardiac troponin. The reviewer suggests minor revision of this manuscript before its publication.
Minor revisions:
The reviewer suggests to remove Table 1 to p7, line 256 as the authors compare the EC-based and FET-based techniques. So, it would be more convenient for the readers to be better understanding the differences of these techniques by putting the comparison at the end of FET section.
We agree with reviewer’s comment, the table 1 is relocated to end of session 3.
2. The concentration description needs to be consistent throughout the manuscript. Either with ng/mL or ngmL-1.
We agree with reviewer’s comment, all unit is unified with ng/mL.
3. A white square in figure-3. Please modify it.
We agree with reviewer’s comment, the white square is removed in fig.3.
4. Regarding the surface plamon resonance based technique for the detection of cardiac troponin I, there are two recent published nice papers to be included in this review article for comparison. (ACS Sens. 2016, 1, 12, 1416-1422; Anal. Chem. 2016, 88, 23, 11924-11930).
We agree with reviewer’s comment, two references added in the SPR session. Please check the reference 83 and 84.
5. The reference section is messy. Missing pages in references: 2, 7, 12, 21, 45, 46, 60, 63, 69, 71, 77, and 88. Some references use journal full name but some are with abbreviation. Please refer to the journal guidelines.
We agree with reviewer’s comment, the following references adapted based on journal guidelines.
6. Typos: p3, line 129 (TNi); p5, 198 ([55Liu]); p6, line 246 (TN I); p9, line 379-380 (concentration units); p10, Table 2, ref 95 (0.76 pg/Ml); p10, 402-403, (concentration and wavelength units).
We agree with reviewer’s comment, the typo errors were corrected with red color.

Reviewer 2 Report
This review provides a comprehensive overview of Troponin detection using various nanostructure-related sensing methods including electrochemistry, immunoassay, SPR, FET, SERS. Though the paper is very comprehensive with very detailed numbers such as the detection limit, the opinions from the authors are not very clear. Since a lot of details in the main text, it is not very enjoyable reading the article, as well as getting the main idea. I have several suggestions regarding the manuscript.
I definitely would like to see the performance of each biosensors, but I am more interested in seeing
1) The improvement/enhancement when coorperating with nanostructure in general
2) The exact function of nanostructure in Troponin detection in different approaches
I would suggest to have a review over the clinical requirement and procedure for AMI diagnosis in terms of Troponin detection, including time, sensitivity, the outcome of Troponin biomarker to the clinical decision, as well as concentration in normal people.
Current state-of-art Troponin detection method used in clinical diagnosis need to be mentioned, in terms of their detection limit, time, accuracy, etc. Some of them are for point-of-care, some of them are for clinical lab. Then we have the comparison of developing techniques to the present ones.
The reported LOD may not be comparible from different papers, as some of them may come from the instrument point of view, which is determined by the instrument sensitivity, while some of them may from clinical/sample tests, which is limited from the tests from blank samples. They need to be more carefully examined.
Less detailed description of the details may help the reader to grasp the main idea quickly. Expansion of the Outlook, and more comments and judgement from the author on each technical approach should be integrated.
Author Response
This review provides a comprehensive overview of Troponin detection using various nanostructure-related sensing methods including electrochemistry, immunoassay, SPR, FET, SERS. Though the paper is very comprehensive with very detailed numbers such as the detection limit, the opinions from the authors are not very clear. Since a lot of details in the main text, it is not very enjoyable reading the article, as well as getting the main idea. I have several suggestions regarding the manuscript.
I definitely would like to see the performance of each biosensors, but I am more interested in seeing
The improvement/enhancement when coorperating with nanostructure in general
We agree with reviewer’s comment, we explained the improvement effect when the nanostructure is introduced to troponin biosensor. Please check the line 108, 278 and 343.
The exact function of nanostructure in Troponin detection in different approaches
We agree with reviewer’s comment, we explained the improvement effect when the nanostructure is introduced to troponin biosensor. Please check the line 180, 278 and 343.
I would suggest to have a review over the clinical requirement and procedure for AMI diagnosis in terms of Troponin detection, including time, sensitivity, the outcome of Troponin biomarker to the clinical decision, as well as concentration in normal people.
This question is out of point for our manuscript intention. We just want to show the nanostructure-based troponin biosensor.
Current state-of-art Troponin detection method used in clinical diagnosis need to be mentioned, in terms of their detection limit, time, accuracy, etc. Some of them are for point-of-care, some of them are for clinical lab. Then we have the comparison of developing techniques to the present ones.
This question is out of point for our manuscript intention. We just want to show the nanostructure-based troponin biosensor as the proto-type test, not for clinical application and point-of-care application for AMI detection in the current stage.
The reported LOD may not be comparible from different papers, as some of them may come from the instrument point of view, which is determined by the instrument sensitivity, while some of them may from clinical/sample tests, which is limited from the tests from blank samples. They need to be more carefully examined.
We compared the LOD of recently published works for troponin detection, please find the table 1 and table 2.
Less detailed description of the details may help the reader to grasp the main idea quickly. Expansion of the Outlook, and more comments and judgement from the author on each technical approach should be integrated.
We agreed to reviewer’s question, we added some comments in the outlook session. Please check the line 458.
